# The Ecological Turn in Design: Adopting a Posthumanist Ethics to Inform Value Sensitive Design

Steven Umbrello 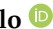

Institute of Ethics and Emerging Technologies (IEET), University of Turin, 10124 Turin, Italy; steven.umbrello@unito.it

**Abstract:** Design for Values (DfV) philosophies are a series of design approaches that aim to incorporate human values into the early phases of technological design to direct innovation into beneficial outcomes. The difficulty and necessity of directing advantageous futures for transformative technologies through the application and adoption of value-based design approaches are apparent. However, questions of *whose* values to design are of critical importance. DfV philosophies typically aim to enrol the stakeholders who may be affected by the emergence of such a technology. However, regardless of which design approach is adopted, all enrolled stakeholders are human ones who propose human values. Contemporary scholarship on metahumanisms, particularly those on posthumanism, have decentred the human from its traditionally privileged position among other forms of life. Arguments that the humanist position is not (and has never been) tenable are persuasive. As such, scholarship has begun to provide a more encompassing ontology for the investigation of nonhuman values. Given the potentially transformative nature of future technologies as relates to the earth and its many assemblages, it is clear that the value investigations of these design approaches fail to account for all relevant stakeholders (i.e., nonhuman animals). This paper has two primary objectives: (1) to argue for the cogency of a posthuman ethics in the design of technologies; and (2) to describe how existing DfV approaches can begin to envision principled and methodological ways of incorporating non-human values into design. To do this, the paper provides a rudimentary outline of what constitutes DfV approaches. It then takes up a unique design approach called Value Sensitive Design (VSD) as an illustrative example. Out of all the other DfV frameworks, VSD most clearly illustrates a principled approach to the integration of values in design.

**Keywords:** value sensitive design; design psychology; posthumanism; applied ethics



## 1. Introduction

Many of the events, interactions, and processes that humans engage in are carried out by means of technological artefacts. Various technological instruments and systems can account for the majority of our highly systemic and interconnected world, becoming critical nodes to the continued fecundity of this infrastructural assemblage. Think of electric power plants, automated shipping vessels, the computer, surveillance cameras at every street intersection and, of course, the smartphone in almost every person's hand. These objects are not some foreground imposed ad hoc on a 'natural' backdrop of nature. Instead, they are enmeshed, heterogeneous symbioses built upon a foundation of other technologies that are at least partially responsible for their emergence. This is the technological milieu.

When we look at the current (and ever progressing) state of technology but ignore the layers of preceding innovations that permitted today's state, it becomes easy to miss the forest for the trees. Fire, gunpowder, steel metallurgy, and the air pump all set the stage for later technologies. This progressivism has made it impossible to discretely separate any technology from its situatedness. The independence of any technology from its context is thus anachronistic and non-representative [1,2]. What we presumably see in the majority of nature-cultures [3] around the world is a co-constitution of technology and the social—hence the term 'sociotechnical systems' [4,5]. This co-construction imposes constraints on

the landscape of potential futures is open to us, delimiting future possibilities. How we design technologies, what design approaches we choose, technical limitations, and how they can interplay and constrain our choices about the future all become valuable questions in the twenty-first century.

The trend towards further progress, 'modernisation', technological incorporation, transformation, and (in some cases) transcendence [6] makes the turn towards design of particular salience. Because of this, what has been termed the 'design turn in applied ethics' has emerged as an interdisciplinary field of study that seeks to make a practical impact on technological design in order to better direct the design and development of artefacts towards desirable futures [7,8].

Various technological catastrophes in the twentieth and twenty-first centuries have provided ample evidence for the need to re-evaluate how technological development is underpinned and eventually made ubiquitous, if at all [9]. Absent responsible research, innovative strategies, and interventions, these incidents will continue into the foreseeable future. The need becomes even more pressing as technological innovations converge, pushing the limits of our moral intuitions [10] to induce novel and difficult-to-deal-with moral overload [11]. In response, design philosophers have envisioned various methodologies and approaches to help incorporate human values into the early phases of design. This way of thinking about design is called a Design for Values (DfV) approach, which typically takes the form of principled and formalised frameworks. Designers can levy these frameworks to account for the values of design agents and enrolled design histories that are implicated by a sociotechnical-institutional innovation under consideration.

Despite these early efforts and their success at embedding values in design, existing DfV approaches fail to account for the position of nonhumans in the technological milieu by relegating them to a static background in discussions. The ecological turn in science and technology studies or STS (advanced by scholars such as Bruno Latour, Donna Haraway, Graham Harman, and Timothy Morton) has made strong arguments for the total interconnectedness of all forms of life, along with the symbiotic and asymmetric relations these entities have with one another. For DfV approaches to be genuinely salient to what has been termed the sixth mass extinction event—mostly due to anthropocentric, technological modernisation—they must account for nonhumans as essential stakeholders in their design considerations.

The purpose of this paper is to show both why and how nonhuman animals are fundamental to all design approaches if designers wish to design artifacts that ontologically map in an authentic way. I employ one DfV framework, the Value Sensitive Design (VSD) method, as a way to demonstrate this incorporation. I selected this method because compared to other options, VSD makes conceptual investigations (i.e., philosophical inquiries) relatively explicit in its methodology. Showing how VSD severs the nonhuman from design along with how this gap can be breached has implications that span the entire design methodology discourse. If this paper is philosophically cogent, DfV approaches can be bolstered against anthropocentric bias as well as become ecologically sensitive.

To the best of my knowledge, this is the first concerted paper to: (1) evaluate the merits of the DfV approach by drawing primarily from ecological literature in STS, and (2) determine how, in light of this ecological thought, nonhuman animals can become design agents during the value consideration stage of VSD. Prior literature on DfV, and VSD in particular, has emphasised the structure of the approach itself [12,13] along with how these methodologies can be applied to both current innovations [14–16] and speculative future innovations [17–19]. However, although this literature provides useful information for understanding DfV approaches and how they can function, it remains thoroughly anthropocentric in its value investigations. This means it fails to account for necessarily broader implications, enrolments, and enmeshments for nonhuman agents. The paper is thus comparatively unique within existing scholarship as it integrates foundational ecological literature central to STS with Italian discourse in the posthumanities that has made significant headway into philosophical inquiries on nonhuman animals. The work

here intends to spark debate and further discussion on the place of nonhuman animals within design discourse.

To tackle these considerable hurdles successfully, the paper is organised as follows. To begin, Section 2 thoroughly outlines the VSD approach as it is characterised in existing literature while emphasising its conceptual investigations. Section 3, which comprises the bulk of the paper, works through ecological literature with an attentive focus on the comparatively unique aspects of Italian discourse and the implications for nonhuman animals. Section 4 provides some initial cursory recommendations for how the values of nonhuman animals can be enrolled into design approaches. Section 5 concludes by outlining the limits of this paper's investigations and areas for potential future research.

## 2. Technological Assessment and Designing for Values

One of the fundamental premises on which the *design turn* is built upon is a shift away from static, adjudicatory accounts of moral deliberations (absolute dilemmas, trolley cases, etc.). Instead, they look towards ways to open up alternative future possibilities by intervening in design or by envisioning a third option [7]. One earlier effort focused on the values inherent to technological innovation can be seen in the practice of *technological assessments* (TA) [20–23]. Forming part of what has been termed the *responsible research and innovation* (RRI) discourse, TA was and is employed as an approach to providing policymakers and governing bodies with tangible cost–benefits for the adoption of specific innovations (rather than more general or encompassing technological artefacts and systems). However, this mode of inquiry, various instantiations, and flavours has been criticised as problematic given concerns for the scarcity of information regarding the consequences of adoption early on along with the ability and necessity for early design intervention. These concerns are coupled with the fact that once technology becomes ubiquitous, directing it becomes incredibly costly—if not impossible [24]. Still, TA provided the RRI discourse with informative tools and avenues that primed parallel approaches towards the necessity for anticipatory design and governance [25–27].

### 2.1. Value Sensitive Design

The crux of VSD is premised on the need to account for human values during the early design phases of a technology [12]. Coined by Batya Friedman, VSD initially emerged from the human–computer interaction (HCI) community to investigate how human interaction with information technologies and networks takes place and how it can be directed via design [28]. Other design approaches within HCI were adapted concurrently with information and communication technologies (ICT), each of which prioritised different methodological tools or procedures. Examples include universal design, inclusive design, participatory design, and worth-centred computing among others [29–32]. Among existing frameworks as well as others that have not be attributed proper names, some scholars [33–35] have argued that the VSD approach is the most encompassing. This is because it possesses the most extensive scope and principled approach to the design of technologies with human values at its core [36–38].[1]

Developed in the last decade of the twentieth century, VSD arguably offers "a theoretically grounded approach to the design of technology that accounts for human values in a principled and comprehensive manner throughout the design process" [28] (p. 70). Although initially used as a broad term for anticipatory approaches to design, Friedman later formalised VSD into a formal framework that, soon after, began to take hold within burgeoning RRI discourse [39]. Although Friedman remains perhaps the most prominent

---

[1] For this reason, the VSD method is taken up as the comparative approach in criticising value-investigations in the *design turn*. Of course, there are other more commonly practiced applied design methods (such as Human Centered Design or HCD) that can be used as a locus to bridge the goals of this work with a larger community of current design practices. HCD or Design thinking is implicitly anthropocentric and very widespread today. Within the design community, designers have proposed adding a fourth (albeit incremental) dimension to HCD's Desirable, Feasible, Viable triage: Responsible. 'Responsible' is meant to be a catch-all dimension that would demand for design to consider its impacts whether unintended or not, such as pollution, killing of animals, climate change, etc. Some have suggested a Bio-centric Design (BCD to replace HCD).

figure in VSD literature, penning multiple journal articles and chapters, the approach has taken hold most firmly in the Netherlands (mainly by scholars at the 4TU.Centre for Ethics and Technology[2] and the similar Delft Design for Values (DDfV) Institute [40,41]).

Given the prominence of 'value' itself within the DfV tradition and VSD in particular, the definition of values is of singular importance. However, Friedman never gives a full account for what 'value' consists of or in. Instead, the authors opt for a more general account of value that is directly implicated within systems design [28] (p. 2). They then proceed to provide a list of enrolled values most relevant to system design, which have since been used as the benchmark for universal applicability of the VSD approach. These values are human welfare, ownership and property, privacy, freedom from bias, universal usability, autonomy, informed consent, accountability, identity, calmness, and environmental sustainability [28] (pp. 17–18).

The latter could serve as an appropriate value for this paper, as it can be construed as a value that maps on to that of the enrolment of nonhuman animals in the design process. However, each of the listed values designated by Friedman et al. are explicitly 'human values' and to this end must primarily serve human needs. If present at all, the 'values' of companion species are ancillary. Thus, environmental sustainability is not the preservation, protection, or enrolment of ecological agents *for their own sake*. Instead, it "refers to sustaining ecosystems such that they meet the needs of the present without compromising future generations [of humans]" [28] (p. 18).

Still, the theoretical foundations of these distilled values aim to address the relational, multi-use nature of technology. VSD affirms that, along the lines argued by Winner (1980), values are designed into technologies. Further, it affirms that the manner in which technologies are implemented and appropriated is a sign of specific values. To that end, the multi-use dimension of any specific technology can implicate a set of unforeseen values during design. This becomes ever truer across various societies and cultures where a multi-set of values emerges. Proponents of VSD methodology argue that any possible technological innovation must be situated within its sociocultural milieu, forfeiting the potentially disastrous goal of universal design [42,43].

At its inception, the VSD scholarship primarily focused on ways to formalise a design approach for adoption by designers in order to account for values in design. Friedman and colleagues divided their principled methodology into three-step or "tripartite" analyses called "investigations"14. The three analyses are (1) conceptual, (2) empirical, and (3) technical. No analysis stands on its own. Instead, they together form what should be a self-reflexive and continually 'iterative' method to account for values during both the early design phases and throughout development [44]. In the following sections, I briefly outline each of the three investigations as conceived by Friedman et al.

### 2.1.1. Conceptual Investigations

Conceptual investigations are typically characterised as the philosophical leg of the VSD stool, marking the initial entry point for the methodology. The primary aim of the investigation is twofold. First, it considers the technology in question in order to identify enrolled stakeholders. Typically, there are two types of stakeholders. *Direct stakeholders* are those individuals or groups who will implement the technology (the users). *Indirect stakeholders* are those who are affected by the proposed artefact, but do not implement it in the first-hand way that direct stakeholders do [13]. The latter become of particular interest when considering the place of nonhumans in a design framework, as I discuss in proceeding sections. To illustrate how a conceptual investigation is carried out, consider speculative intelligent agent technologies such as lethal autonomous weapons (LAWS). Conceptual investigations could allocate direct stakeholdership to both the soldiers implementing the LAWS and those identified as enemy combatants. Indirect stakeholdership could be

---

2    The '4TU' represents four technology-focused universities in mutual collaboration with one another. They are Delft University of Technology, Eindhoven University of Technology, University of Twente, and Wageningen University.

attributed to non-combatants who may be caught in the crossfire without any clear way to defend themselves. It could also be attributed to the designers of LAWS themselves, who are linked to notions of liability and responsibility even without direct involvement in the use of the technologies in the field (Umbrello and De Bellis, 2018 discuss similar examples of intelligent agent technology under the VSD framework).

Second, the conceptual investigation looks at the potential values of those groups enrolled in the relevant innovation. This aim is critical for determining stakeholders. For example, a conceptual review of advanced (mostly speculative) nanopharmacy technologies implicates a need to contextually determine values such as safety and efficacy. Safety is already discussed and regulated under nonmaleficence[3] in the decision-making process of medical practitioners [17]. It is during this phase of analysis in a conceptual inquiry that value conflicts arise. Recourse to existing philosophical literature is typically recommended to 'weigh' values and guide designers towards more salient avenues [11]. The moral overload of values that may come into tension are not characterised as strong disjunctive dilemmas, but instead as restrictions that may exist within design space. Potential design solutions can come to bear in order to increase the salience of a design *flow,* which better accounts for initially conflicting values [45].

### 2.1.2. Empirical Investigations

Empirical investigations typically involve the direct enrolment of identified stakeholder groups to evaluate their unique situatedness, their "understandings, contexts, and experiences of the people affected by the technological designs" [13] (p. 1251). Scholars have proposed various methods for doing so. These mostly draw from ethnographic and social-scientific toolkits, which allow designers to co-create and collect data that best reflects the desires of relevant groups. This can be done through the employment of participatory observation, survey, direct interviews, or even the use of *Envisioning Cards* [46]. The values defined through initial conceptual investigations can then be revised and redefined during empirical elicitation. Similar empirical work is done with the stakeholder agents themselves. During this stage, "issues of technical, cognitive, and physical competency of stakeholders" are scrutinised to determine whether relevant stakeholder values have been sufficiently evaluated (typically through elicitations of group representatives [45]) (p. 97).

### 2.1.3. Technical Investigations

The discrete function of this investigation is to look at the relevant technology itself. This helps formulate questions about how technical implementation of the technology implicates the values investigated beforehand, as well as about the restrictions and constraints imposed by values within contextual scenarios. This brings into question the effect of the material constraints of the technical system on value tensions. Because emerging values are often enrolled once technologies have left the design space to become ubiquitous, architectural imperatives for flexibility are likewise enrolled. Technical investigations thus often look for ways to design system architectures. These ways should be receptive to emerging values over time and in different contexts [47,48].

In sum, VSD emerged in parallel to various other design methodologies that aspire to achieve value sensitivity through design considerations. It was founded and developed as a principled way to account for the sociocultural and ethical implications of HCI. VSD emphasises co-interactions across various (direct and indirect) stakeholder groups through information technologies, systems, and artefacts. This emphasis enables realisation of its central premise: the operationalisation and strengthening of human values through technological design [12].

Although approaches to VSD differ in their sociocultural context and the particular artefact in question, they commonly adhere to an initial investigation of the conceptual values implicated by a particular design in context, including how those values can be

---

3　This is only one of a set of values that is context-specific to 'care', as broadly construed within the medical domain.

defined, refined, and actualised in early design phases [49]. Values deemed pertinent to design are rendered as an a priori-operative necessity for design. This means they are defined in terms of functionality and as technically implementable by designers, rather than as abstract concepts. Then, empirical investigations determine the tensions and applicability of those values in design. If necessary, they reiterate those observations back into conceptual investigations for reformulation and definition.

The reiterative nature of VSD makes it a non-absolutist design framework. Values are conceptualised, evaluated, and implemented into design. Trials can be conducted to determine whether the design meets value-specifications according to stakeholder desires. If not, designers can return to tripartite investigations and make different modifications to arrive at a more salient product. These short, evaluative loops allow for continual changes and modifications that can, in turn, accommodate intermittent societal change (as opposed to the rapid, often revolutionary changes that transformative technologies can ultimately induce).

Therefore, far, I have provided a cursory account of the history, motives, and function of DfV frameworks with a focus on the VSD method and its explicit philosophical implications. The remainder of this paper presents a discussion of what Timothy Morton calls 'ecological thought', how this posthumanist position on the necessary interconnectivity of the biosphere has severe implications for technological design, and how Italian scholarship on this particular subject provides some critical insights that can inform VSD—and DfV approaches, by extension.

## 3. Bridging a Severing of Praxis

Before any serious attempt to include nonhuman animals into a thoroughly anthropocentric domain, it is best to clearly delimit the aims of this paper and explicitly state its limitations. VSD arose in (and still bears a robust umbilical connection to) the unquestionable analytic tradition of philosophy. The literature that designers *ought* to consult throughout their value investigations is assuredly an ethical one falling within the arena of post-Enlightenment—mostly post-Kantian—thought, which characterises the analytic tradition as a whole; i.e., living in Kant's shadow.

In line with the above intellectual tradition, then, the reader should note this paper is not merely a 'potential path for nonhuman values in design'. The method by which this goal is articulated implicates far more. The very contention of drawing from a posthumanist tradition towards what has traditionally been an analytic praxis[4] [10] plays a tumultuous game of trying to bridge the severing of the analytic and continental traditions of philosophy that, if anything, have been separating at a greater pace [50]. Although this paper does not aim to explicitly achieve some homogenous unification of two discrete philosophical traditions, it nonetheless implicates an asymmetric symbiosis between facets of the two by engaging in this very project. It necessarily follows that any single work pursuing such a goal should be met with a critical gaze, mainly when such implications are at stake. Any practical success, applicability, or usability of the philosophical insights provided by this paper can only be determined a posteriori. Design is by nature an applied practice that, despite having conceptual parts, ultimately and necessarily moves into the realm of the real.

To take on what traditionally have been marginalised actors (nonhumans, other forms-of-life [51], or significant otherness [3]), we need a thorough overview of the philosophical foundations from which these issues emerged most potently—that is, of posthumanism itself. Because this paper draws from this thought-tradition as the crux of its thesis, a distilled account would only be marginally beneficial. Although no single paper could provide a total, conclusive account (indeed, totality and absolutism are not endorsed here), a rounded explication of posthumanism and its Italian flavour is certainly warranted.

---

[4] In [10] author discusses the applicability of intersubjectivity (a primarily continentally-derived notion) towards DfV approaches. However, this is the exception and not the rule.

### 3.1. Posthumanism and Reclaiming Animality

Posthumanism is a strange, multifaceted, and difficult-to-atomise collection of thought. This is part and parcel due to its obscure origins and the variety of ways that it is used and defined [52]. Common to this hermeneutic plurality of posthumanisms is the notion that the central, privileged place of humans—the heritage of theological and philosophical history—is untenable, if not illusory [53,54].

Various fields, such as critical studies, philosophy, anthropology, and sociology among others, have appropriated the term for various goals and with differing interpretations. Although often confusing and clouded in jargon and obscurantism, the theoretical foundations of posthumanism (and particularly that of contemporary posthumanism [55]) encompass what has been called 'ecological thinking' and the ground-breaking philosophical move of flattening ontology [56–58]. What exactly does this mean? It means that 'humanity' *in itself* is a fragile concept, or even one that is illusory, built upon false notions of the necessity of human cognitive superiority [53]. Once humanity is decentred from both the centre of the universe and the biosphere, then the special place that humanity endowed itself with becomes tender and easily bruised [55].

Ecological thinking becomes the natural consequence of this decentring, this movement towards the fringes of thought [57]. As a result of the dissolution of the human ontological pedestal, nonhuman animals and other forms of life come into the fold on an equal ontological basis. Speciesism becomes nothing other than a tool of economy that drives humanistic conceits embedded in hypocritical infrastructures and techniques. It is thus an ethical choice. Inconvenient truths concerning the absence of reasons to massacre millions of animals become apparent and imminent [59–61]. It follows, then, that the most authentic starting point for ontology is a flattened one in which animality is equal for all life forms. It is not one that begins with human superiority (i.e., the tradition of theology and humanism). This implicates the need to reclaim animality [62].

Given that the philosophical focus of this paper is on the ontology of nonhumans in design, Italian posthumanist literature naturally comes to the fore. As mentioned, posthumanism is a varied set of concepts with origins across disciplines that speak to different uses. What makes the Italian tradition unique is that the few (self-described) posthumanist scholars arose from within discourse on animal ethics, making the Italian flavour of posthumanism particularly attuned to thinking about nonhumans. Hence, the works of Leonardo Caffo and Roberto Marchesini become of primary utility here. Both publish primarily on posthumanist intersections with animal studies. The following arguments employed for this reclamation of animality are thus utilised to inform the salience of ecologically conscious design.

### 3.1.1. Metamorphosis

Within the classical humanist tradition handed down since Plato and made most explicit through theological institutions and philosophies, animals *as such* do not 'exist'. In a similar vein was Heidegger, whose philosophy was famously built on the forgottenness of *being*; ironically, he excluded the *being* of other forms of life that were not Dasein or not German-Dasein [50]. The being of others (nonhumans)—*other-being*—has been traditionally marginalised. If not, it is merely regarded as unsubstantiated according to the inherited 'Great Chain of Being' of theology, which entitles the immateriality of the human soul as the object of ontic privilege. This is what the Italian scholar of posthumanism, ontology, art, and architecture Leonardo Caffo calls *speciesism*. Speciesism is the "engine of economy" and a tool of 'concealment' used to *de-being* (my neologism) other forms of life, thus making their use as a tool (as food) more palatable [55] (p. 9).

As speciesism becomes a part of ethical nomenclature, so the reasons employed to support the massacre and continued slaughter of millions of animals become necessarily void. A new starting point must be induced, one that begins with equal animality. Rather than affirming the privileged place of humans/human-being *a priori*, this new starting point permits a more authentic ontological onset. Consequently, *anti*-speciesism runs contrary to

the anthropocentrism of humanism and contributes to the growth of posthumanism [55] (p. 20). It should be clarified that 'anti-speciesism' as such is not a negation of 'species', but that of the 'ism' of exception. Anti-speciesism can nonetheless be taken as a form of power politics used to oppressively flatten the ontologies of various forms of life and the plurality of un-totalisable phenomenologies. What we need instead is a weak form of anti-speciesism that acknowledges the heterogeneity of beings, envisioning open futures for possible worlds other than the one under criticism at any moment. This enables the emergence of a larger 'togetherness' over a 'sameness'. Togetherness ferments an intersubjective respect rather than the projection of uniform qualities[5] [3].

Metamorphosis towards a new social ontology becomes necessary. Nonhuman animals are not objects or automata, as Descartes suggests, that have ontic status merely as the objects of gaze [62,63]. The specious notion of the human/animal dichotomy is a constructed historical phenomenon; the semantics of animality have been similarly used as a socio-historical tool for discrimination [55,61,64]. Modern developments in zooanthropology have clearly observed that nonhuman animals are capable of complex social interaction and phenomena [65–67]. Social ontology, then, must bridge a significant gap between social studies of animals and the inherited philosophical ontology of animals [68].

### 3.1.2. Metaphysics

The privileged position of the human-*being* has been historically and philosophically assumed across multiple schools of thought. Ptolemaic cosmology (geocentrism), for example, is a particularly erudite metaphor for this entitlement, i.e., 'man as the centre of the universe' or, more to the bone, 'man as the centre of ontology' [55] (p. 27). Naturalistic discoveries showed that geocentrism was false and advanced a heliocentric view. Posthumanist thinking usually assumes that the stories at the foundation of a geocentric view are consequentially false[6].

Similarly, the later Kantian conception of the *transcendental* is even more devastatingly anthropocentric. This critique of Kantianism (and the post-Kantian philosophy that dominates the Anglo-American analytic tradition) is one of the standard features of posthumanism. More broadly, it is a feature of the Speculative Realist movement gaining steam in modern continental philosophy [50,69,70]. This critique of Kant explicitly aims at what Meillassoux (2009) called *correlationism*, which suggests that what exists is solely the correlate between subject and the object (with the subject always the human). Although Kant is generally understood as being correct that direct epistemic access to *das Ding an Sich* is impossible, all of the speculative realists and their posthumanist cousins agree that Kant was not Kantian enough. They mean he did not push his philosophy to its logical end[7]: that object-object correlations exist, that the human subject may have no direct access to any other object (the illusive *thing-in-itself*), and that other objects may have no direct access to humans. This mode of ontology naturally confers a strong form of realism, beginning on an equal ontological footing rather than *a priori* attributing the correlational power of reality to just one particular type of object (i.e., humans).

By a similar token, Kant's partially finished project is a radical philosophy that is philosophically geocentric. The centre of the circle of reality in which humans have placed themselves is the very circle they created by/for our own reason (Caffo, 2017, p. 30). To some extent, the implications of this philosophical inheritance have been acknowledged in the discourses of sustainability and the need to protect and ameliorate the earth. However, sustainability discourse (sustaining what exactly?) only extends so far in this direction as it

---

5   Haraway [3] provides a good way to envision this togetherness in terms of various nature-cultures between humans and dogs. Togetherness would be based on finding common respect for difference rather than anthropocentrically anthropomorphising behaviour and projecting anthropic consciousness to nonhuman beings as a way to relate.

6   Such a philosophical move is based on questionable logic and borders on committing the genetic fallacy. Still, the theological and purely coincidental observations of pre-moderns can be criticised on their own grounds, which I contend are similarly false.

7   Like Heidegger's claim of Dasein being particularly human, and mainly German at that. Making Dasein a feature of all lifeforms would begin on a more egalitarian ontological footing.

does not jeopardise the stability of the financial elite or the anthropocentric privilege that we created and gifted ourselves [71].[8]

Therefore, then, what is the consequence of heliocentrism as a historical development? The Copernican revolution resulted in a consciousness shift that moved the human from the cosmological (ontological) centre to the periphery; Earth became one planet among others in a vast universe. Philosophically, then, the anthropocentrism that was packaged with geocentrism must be similarly abandoned. This privileged space of centrality must be discarded as it does not exist. With this abandonment, everything can begin from a more philosophically genuine position of equal ontological footing in the periphery. The sensation of estrangement becomes the philosophical foundation for posthumanism. As the centre becomes vacuous and the periphery crowded, ecological thought is born. Such thought is based on acknowledgement of the common substances that constitute biodiversity and a reconceptualisation of non-anthropocentric space [55,72].

All things then exist on a necessarily connected, *enmeshed* [58] ecosystem composed of inexhaustible objects that are greater than the whole. The parts (objects, entities, actors, etc.) are *lifeworlds* in themselves, inexhaustible either through a literal definition of atomisation or by their effects on other objects [56]. The depth of their being is never directly accessed, but always remains veiled and more than what it appears.

Posthumanism can thus be understood as a 'rebirth'. Our substantial form must be altered as a means to conquer the anxiety that comes from the geocentric-heliocentric shift, both in its cosmological and philosophical sense. The movement from centre to the periphery is an unquestionably dangerous shift, given the infrastructures, networks, and assemblies inherited and built upon that substantiate what has been argued to be an illusory conception of the 'human' and its place in the world.

Contra to Heideggerian metaphysics, we can therefore alter our forms. We humans are not possessed by *being* as such. This consists of the abandonment of Western absolutism for a new role, a role *among many* towards the affirmation of animality. On the periphery, our roles continuously shift as we tangle with other beings. We should not aim for the creation of a new circle or centre in the periphery, but to understand that all beings are infinite in their enmeshments [3,73]. The conception of the cyborg and the centre-periphery shift makes anthropocentrism the emergent quality of a form of 'local anthropocentrism', more specifically. This specific form of anthropocentrism has privileged not just all humans, but a subset of human 'types' in particular: the male human who is white, Western, and heterosexual. On the periphery, the unveiling of *being-as-animality* becomes clearer whenever these categories change.

Once we shift from the centre, we see that this metaphysics entails an ethics—not only of the 'human as the ideal', but the 'ideal of human'. Both are symbolic vestiges of an excessive and exceeded past.

### 3.1.3. Materialism

Theological creationism, particularly that of Western proclivity, is substantiated by narratives of (the Judeo-Christian) God creating 'man' to dominate other forms of life:

> Then, God said, *"Let us make humankind in our image, according to our likeness; and let them have dominion over the fish of the sea, and over the birds of the air, and over the cattle, and over all the wild animals of the earth, and over every creeping thing that creeps upon the earth."* (Genesis 1:26)

The emphasis here is on verticality, on dominance over divine creation and the ontological entitlement of one form of being over others. This tradition has been the heritage of Western philosophy and humanism, as such. This God/human dichotomy, which is almost univocally rejected by posthumanism (alongside all dichotomies and bifurcations),

---

8  Sustainability discourse has been quite egregious given its association with renewable energies. Counter-literature has discussed how discourse on sustainability is directed at merely sustaining currently destructive practices. These practices may consequentially lead to the continued and unmitigated use of environmentally devastating forms of energy and fuel development.

has been addressed through various philosophical recourse. Nietzschean power politics are one such recourse, as well as the Sartrean approach to understanding liberty. All of this is done in a way to evaluate an inherently absent divinity that embroiled the angst of human self-importance.

However, as with the Copernican revolution, Darwinism revaluated the hierarchy of being to move away from the top-down 'great chain of being' with God at the pinnacle and humans slightly below (dominating over the 'lesser' lifeforms gifted to them). Instead, Darwin revealed a bottom-up progression of lifeforms made up of the same substances. He showed that humanity—and all other life—is the product of a chaotic process rather than a divinely ordered one. Because there is no ontologically superior substance of the 'soul' bestowed upon humans, the divine *telos* is dissolved. What Darwinism does, then, is open up philosophy to what is outside humanity. A metaphysics of ecology, of monogenesis, is born from Darwinian material investigations [74,75]. The fundamental philosophical tool unveiled by Darwinism is this opening up to animality.

In sum, contemporary posthumanism exposes an ontological continuity based on monogenesis to other forms of life. This proposes that individuals live in a perpetual state of anticipation, in the present, concerning the things external to them [55] (p. 56). It applies to the individual human and thus orients the singleton to those like themselves (in this case, all beings). From here, a new form of ethics can be envisioned.

### 3.1.4. A Companion Ethics

What does this entail for ethics as it exists? The current Moral Law Theory tradition of moral theorising (utilitarianism, deontology, and offshoots) are post-Enlightenment inheritances based on humanistic vestiges [76,77].[9] Such anthropocentric morality must be abandoned. Our new habitat on the fringes holds a damaged planet, one in ecological crisis, in clear sight [57]. This truth becomes remarkably apparent when a species (humans) radically destabilise their own survival. The crisis provides the fermentation needed to birth the posthumanist. Body-Oriented ethics becomes the most obvious here, wherein the subject does not exceed the limits and rights of the body of other without consent (except when excessiveness is necessary as in self-preservation) [55] (p. 62).[10]

Similar to the philosophies of Singer and Deleuze, the body becomes an inviolable boundary for moral actions [78–80]. The environment that houses humans is in crisis as a result of an ethics blind to the bodies of others. A posthumanist understanding of this does not discriminate against sex, ethnicity, or individual preferences. Being-in-the-world has always naturally implicated a certain level of violence. As such, this implicates all life forms as a part of a single painting—an assemblage of systems and nodes.

Body-Oriented ethics is the adaptive mechanism (to use evolutionary terminology) that posthumanism levies to *be-in-the-world* with other forms of life, healing a damaged planet. Whereas transhumanism (what Fuller [53] calls *ultra*-humanism) conceptualises human limits as a resource, posthumanism argues that humans do not have to become functionally immortal to continue as they do today. Instead, humans must learn that the posthumanist is not something that has to be consciously brought into being (i.e., with a technofix [81]). Humans were always already posthuman.

This philosophical ethics requires a theory of anticipation that allows the posthuman to develop and take hold. As a merger of art and architecture, design that takes this posthumanist ontology into account is primed as a candidate for a newly formulated ethical theory that is non-anthropocentric.

---

9　These moral law theories should be distinguished from other moral traditions like virtue ethics, which have deep and ancient roots.

10　This is notably different from the deontological ethics prescribed Kant, as it is not so oppressively universal or absolutist. Instead, it is deeply contingent on context.

## 4. Designing Anticipation

If we are to affirm a realism concerning objects, anthropocentrism must be subjected to crisis[11]. This realism is the consequence of the dissolution of anthropocentrism and the Kantian (exclusively human) subject-object correlation. Object-object relations correlate each other independently of human consciousness. Accordingly, every form of life perceives the world in its own way. There is a plurality of inexhaustible phenomenologies [50,82,83]. Hermeneutics exists in relation to a single, real world that exists and becomes available to interpretation.

Philosophy thus provides the initial ontological landscape that can then be built upon. However, alone, it is insufficient; what is necessary is the marriage of ontology with architecture and art. A theory of anticipation, built viz. these three, can be used to form the structure for posthuman to grow in [55,84]. This means posthumans[12] must *live their environment* rather than *in* it, abandoning the post-phenomenological conception of 'nature' as a static background for human action [85]. Along the lines of Haraway's *Staying with the Trouble* (2016), posthumans must seek to live in the places left abandoned and desolated by excessive human consumption [55,86] (see also [79] (p. 71). The wastelands of capitalism can provide a fruitful breeding ground where posthumans and their varied between-species relations can flourish [55,87,88]. Art and architecture poise themselves as the best candidates for giving a substantive praxis towards the actualisation of posthumanist theory. What architecture does is graft onto the foundations of mutual, co-habited spaces without destroying, protecting forms of life without isolating them from one another [55] (p. 75).

Doing this allows for the construction of new spaces of hybridisation. The hybrid co-constructed with companion species is not simply reducible to the sum of its parts. Hybrids should be understood metaphorically, rather than literally (although the latter is a speculative possibility); hybrids are an instrument for understanding that forms of life are nuanced, interconnected, and not easily demarcated by clear boundaries if possible at all. This hybridisation should be understood in the terms offered by the evolutionary biologist Richard Dawkins, wherein all living entities exist along a continuum with all other entities [89][13].

What architecture does for the posthuman is aid in preparing them to live in micro-communities, to live in accordance and mutual respect with nature, without discriminations of moral types or the creation of substance hierarchies of dominance. Roles are instead assigned to actors according to their proclivities, desires, and competencies [55] (p. 97). This should not be confused with the grand, universal narratives of utopia that have often been rejected by post-modern thought [90,91]. Instead, the question of how to 'become better' will prove enduring as inter-species communities are *ipso facto* dynamic.

This dynamism requires anticipation as its key towards relative stability. *Ex abrupto* changes at a substantive level could prove disastrous for these precarious communities [87]. Instead, they call for semiotic changes and the potentiality of their *relata*. This should not be construed as a call for revolution, but rather a philosophical shift towards being exemplary. We can exemplify what is to be a new species (a heterogeneous hybrid of beings) that is continually looking for ways to survive a damaged planet [55,71,84,86].

## 5. Designing Technofutures

At this point, it is worth revisiting what this paper has done so far and perhaps even draw some initial conclusions before moving forward. The first part of this paper has provided a literature review of DFV methodologies, highlighting the VSD approach

---

11   Objects should not be confused with things being 'objectified'; an object can be any entity, event, etc. including nonhuman animals.

12   The term refers to the always-already posthumans of philosophical posthumanism, and not to the other interpretations of posthumans such as that envisioned in transhumanism.

13   More literal conceptions of hybridisation have retarded the progress of philosophical posthumanism recounted herein. This is primarily caused by the transhumanist (i.e., *ultra*-humanist) domain in which the concept of hybridisation has arisen. The will towards progress and human betterment has infected the discourse. To this end, a more metaphorical definition of hybridisation has greater utility in understanding human-nonhuman relations and *being-with* other forms of life.

in particular because of its erudite emphasis on consulting philosophical literature and operationalisation in light of the ethical theories distilled from those investigations. The proceeding section took a starkly diverse approach, both stylistically and philosophically. Whereas the former could be said to be a traditional analytic analysis, the latter is markedly continental in origin and execution.

Some have argued that the analytic/continental severing is illusory or merely a social construct, as if the latter had no bearing on reality. Still, one fact remains: in practice, at least, what constitutes the two (scholars, literature, thought) remains clearly demarcated. In many cases, this is ignored by the other tradition [92,93]. As suggested at the opening of the previous sub-section, this paper contends that a bridging of praxis is needed. While potent, adoption of the analytic modes of technological innovation lacks a fundamental ecological consciousness [94]—the very same consciousness that the contemporary continental tradition strongly argues maps directly on to reality [57].

Therefore, how can this be done? Firstly, it should be noted (both due to constraints on this paper and the overall project) that this paper does not propose any form of totality, homogeneity, or clearly demarcated principled method—nor could it ever. The proposal of a clear set of design rules descends purely within an analytic axiom, which is something that posthumanism does not affirm. It necessarily follows that a hybridised approach is necessary, at least initially to satisfy functional imperatives for a concurrent transformation in the development of technology.

### 5.1. Constructing a Hybrid Project

To reiterate, philosophy, art, and architecture (together forming a hybrid) are the necessary tripartite motor for change and transformation. What the hybrid permits is an envisioning of past errors not as mere static consequences, unchanging, and necessary. Rather, errors are contingent, embodied, and causal. Past errors are continually present and affective, influencing both the present and the future. As the RRI discourse suggests, the scaffolding of earlier technologies limits and constrains the possibilities of open futures for current or forthcoming innovations built upon them [7,95,96]. This entails a reflexive element when envisioning future innovations also; imaginaries and design spaces not only shape the future but affect the present and the hermeneutic frameworks applied to the past [97,98].

The *sixth mass extinction event*, anthropogenic climate change, suggests the need to shift away from current destructive paths and envision new modes of being. These new modes involve more than simply how diverse peoples and groups can come together. They involve the construction of a *third space* between peoples (human and nonhuman) [84] (p. 58). Some additional necessities arise from the formation of this *third space*. It must promote a dialogue that is:

1. Explicit
2. Public
3. Accessible to all interested/relevant stakeholders

Part of what VSD can and should do is motivate the construction of this third space as one that can also operate as a design space. Actually doing so does not come without difficulty or methodological manipulation. However, the founders intended VSD to be flexible for seamless integration into the design context for its adoption. Here, are six guidelines for a *soft universalism* that DfV frameworks, in general, should adopt as starting premises. They are as follows:

1. In order to accurately envision co-habitable, symbiotic futures with nonhuman beings, it is necessary to discard moral law theories of morality and adopt an embodied ethics. This ethics provides an anticipatory landscape for coming to a greater understanding and respect for identities and relations with other entities. In turn, this creates a more accurate mapping of the cognitive undertaking within human moral deliberation. It also allows for intersubjectivity between peoples to take place [99–103].

2. Hybridisation and a realist ontology adduce the decoupling of humanity from its traditional theological and humanistic centre [56,57]. All *being* resides on the peripheries, so this is where transformation, hybridisation, and a space of respect and symbioses can flourish. Shared space permits the distribution of agency and an area for mutual co-habitation [72,84,86,104]. A distribution of agency towards an ecological (rather than an individualist) definition of identity is ontologically warranted, going above and beyond the illusory ones still held and passed down as part of the heritage of theology and agricultural-age societies [58,71,84].

3. Unlike Latour's oppressive ANT of immanence, these third spaces acknowledge necessarily contingent connections between all entities. Such spaces further affirm their discrete heterogeneity [56]. Hybridisation is *ipso facto* a constitution of heterogeneity; as such, it represents the distinct yet enmeshed entities of which it is composed [73,84].

4. An *imaginative rationality* becomes necessary for envisioning the broadest set of possible in order to sanction organic changes taking place as needed. This consists of curating how futures are imagined but without authoritarianism taking hold [105]. Curation should be restricted insofar as the constraints of an embodied ethics.

5. Design should be a collaborative, collective, and shared set of practices and instruments. These should promote the distribution of agency and respect for a shared space [57,84,106]. Forces of domination or hierarchy should be rejected to allow symbioses to occur from any agent.

6. Past papers must be understood as symbiotic; they exert asymmetric effects that impact the present and future. The ontic structures upon which past projects are built call into question what actions are undertaken today for the future. They further interrogate the framework within which those actions are undertaken, i.e., how they constrain and promote specific categories of actions.

If we take into account some of these basic framings towards a posthumanist design landscape, we might wonder what exactly is implicated by design. Design implies *ecognosis* ('eco' for ecological and 'gnosis' deriving from the Greek for 'knowledge') as a necessary set of contingencies for ontology [71]. It is primarily an attunement to the necessary enmeshment of nonhumans to humans, both biologically and at the cornerstone of cognition and imaginative reasoning.

Similarly, it implies the dissolution of the aesthetics/design dichotomy (i.e., appearance/beauty vs. function/utility) that is typically sidelined. This dissolution leaves not only the function of the object unmolested, but also its potentiality. The consequences of the marginalisation of aesthetic beauty are evident in modern architecture design, which favours utility over appearance as if one were independent and contrary to the other. This separation is illusory or untenable, and the decision to force this separation becomes one of ethical importance [107]. It is forced in decisions to allow airplanes to filter dirty air from the front (for those willing to pay more) on towards cheaper seats in the back [108], to permit clay roof tiles that enable city birds to nest and remain safe from restricting elements [109], or to authorise ICT devices to produce 'electrosmog' that disorientates and harms bees [110]. There is no longer any *over yonder* where our waste and pollution can be sent without a reflexive feedback effect [57,71]. This is the consequence of ecognosis or knowing that all beings inhabit a single, real world composed of interconnections that never make up a unified, definable whole.

This perpetual intersubjectivity involves the notion that design can never be unified or absolute—nor has it ever been. There exists no totality of interconnections that can be demarcated, one that is explicitly and distinctly larger than the sum of its parts (as both materialists and idealists are wont to argue). Intersubjectivity and the contingent relations that necessarily assemble agents means the selection of design constrains and restricts possible futures. In searching for a cure to the ZIKA virus, we make an ethical choice to design futures that favour humans over the organism. Here, perhaps we should. By a similar example including sheep, we naturally exclude that which would endanger the inclusion (sheep-killing diseases, foxes, etc.). The ethical/political choices that constrain the

open-future possibilities here are biocentric in their privileging. Yet, arguments nonetheless could be warranted to design along those lines. Design has to account not for movement towards perfection, but towards the best possible futures that can be envisioned. Design must continually remain flexible to the future avenues for design that emerge and surface. The nonhuman can no longer be marginalised as merely an 'other' whose being is markedly independent of that of *Dasein*; this was a false starter that was never ontic.

### 5.2. Attuning to Representation

The previous sub-section provided some rough guidelines, premises, and a generally more authentic mindset that designers should employ when seeking to create and enter a design space. A *prima facie* response could claim that those premises lack any operational-isation factor, making any principled way of formally integrating them difficult (if not impossible). Moving towards best possible futures would assume a holism in design, one that is universally applicable and salient. However, I argue that such a holism is illusory and any past argument for it is unfounded. Instead, salient design should shift away from the inherited anthropocentrism that has dominated technological innovation—the centrism that has caused unprecedented ecological devastation and continues to do so under an erroneous conception of the privilege of human-*being* and the necessary superiority of the constitution of the cerebellum [53,111].[14]

The inception of the Anthropocene as a geological era wherein the anthropic impact on the earth's biosphere has become observable. Part of the geological strata should shift attention towards the anthropogenic, asymmetric symbioses that humans exert on the biosphere. Still, this paper concerns the design-for-*beings* in a way that affirms the dynamism of our assemblages. I have provided at least six framing principles to lead towards a more authentic design space. However, there exists at least one initial (albeit ad hoc) step in that ongoing DfV projects can align their design investigations along this posthumanist framework. One such avenue for this alignment arises out of a marriage of moral philosophy with animal ethics, which is called contractualism. The following section aims to restructure contractualism as a path towards the ecologically aware design that has been already discussed. It introduces the theory, criticises where it apparently goes awry, discusses what is salvageable, and describes how to integrate it with what has been presented thus far.

### Contracting Nonhumans

Contractualism may be used to refer to a variety of positions regarding agents who come into a 'contract' or agreement with one another. However, use of the term here derives from the moral philosophy of T. M. Scanlon, which was originally presented in his opus *What We Owe to Each Other* [112]. Scanlon's conception of contract/agreement is narrowly defined. As opposed to the broader connotations it could imply, he goes to the heart of contractualism:

> *An act is wrong if its performance under the circumstances would be disallowed by any set of principles for the general regulation of behaviour that no one could reasonably reject as a basis for informed, unforced, general agreement* [112] (p. 153).

Thus the focus of Scanlon's moral predicate is on wrongness, not correct or good action. What is correct is merely the negation of what is wrong; what is wrong is merely what is not justifiable or disagreeable to contracting agents. The crux of determining the predicate limits of wrongness or agreeableness rests in the human ability to adjudicate what cannot be rationally rejected. The emphasis here, and within contractualism in general, is on rationality as the primary tool of moral deliberation (i.e., of entering into and consolidating a contract). However, as this paper has already assessed, the reason/desire, reason/imagination dichotomies are vestiges of an anthropocentric theology and faculty

---

[14] The latter is a reference to the necessary superiority of the human rational mind, as argued by theology and Enlightenment philosophy. This contrasts with the notion of the mind as a product of evolutionary contingency, as argued by Darwin.

psychology that formed with the foundations of pre/post-enlightenment thinking. These dichotomies are illusory.

Given the explicit weight placed on the faculty of reason when entering into a contract, Scanlon and others make a clear case against the attribution of the moral status of humans to nonhuman animals [113,114]. This is because nonhuman animals lack the faculty for reason to the degree possessed by humans (a contentious argument). Therefore, they cannot come into contract with themselves or humans in general. However, due to the entitlement afforded to reason in contractualism, the moral theory nonetheless gained popularity within animal ethics discourse as a potential route for integrating nonhuman animals into the moral landscape and granting them equal moral status.

Paradoxically, it is the overtly anthropocentric core of contractualism that undermines it to allow movement towards nonhuman enrolment. As it stands, the theory accounts for human actors as the sole proprietors of moral standing. This extends to all humans, regardless of their cognitive or physical states. It necessarily follows that infant children, as well as those with cognitive impairments, are agents with the full moral status of all humans. However, due to their developmental or cognitive states, those agents lack the very faculty necessary to qualify as a contracting agent.

Any moral theory worth its salt must afford, at least intuitively, moral status to the most vulnerable members of society. The logical rejection of these members would render contractualism untenable as a moral theory that could ever be put into practice. Still, supporters rightfully argue that the consequence of not enrolling these members would stress our moral intuitions to such an extent as to induce societal upheaval and destabilise the moral landscape [113]. The proposed solution is the concept of a 'trustee' or representative (rational human), who can represent the interests of those actors and extend full moral status onto them by proxy. The point here is that regardless of the consequences of failing to engage such actors (and the rejection of nonhumans could lead to similar upheaval)[15], the extension of full moral status is based on the notion of their existence as members of the same species. It is not based on the reason-faculty that substantiates the theory ab initio.

Still, contractualism's emphasis on wrongness as the predicate for moral agreement is one of the strengths that makes it preferable to other ethical theories such as utilitarianism (despite the latter's inclusion of animals). This is because the conception of wrongness in contractualism permits trustees to determine what is *ipso facto* owed to animals, rather than the suffering predicated by utilitarianism [112,115,116] (see also [104] p. 183). Contractualism does not, nor was it ever designed to, provide an absolutist or complete theory of morality. Scanlon himself claimed to have only scratched the surface of what contractualism could be. However, if we reject the anthropocentrism this paper has argued is illusory, then that of contractualism similarly dissolves. This would make membership in the *Homo sapiens* species arbitrary at best, and the extension of the consequentially self-affirmed moral status follows suit. The interconnectedness of the biosphere, mutual symbioses, and the asymmetric effects of object-object relations makes the exclusion of nonhuman animals from moral theorising an archaic, anthropocentric delusion.

What design can do is begin its conceptual investigations with the tool of the trustee in mind, as part of the formal repertoire of the toolbox. When constructing stakeholder analyses to determine who relevant groups are, VSD designers should stray outside the anthropic box to which design is traditionally relegated. This necessarily involves designated specialists taking responsibility for representing the potential interests of the biosphere outside the correlationist circle, that of the human. It could mean determining the potentially deleterious effects that a technological innovation may have on the environment—not for the sake of sustaining human progress, but for the sake of the environment overall.

---

15 Imagine the state mandates the euthanasia of all pets/animals within the *polis*. The companionship that exists intersubjectively between species, particularly human-nonhuman, could foreseeably result in a similar destabilisation of society.

The flexibility and reflexivity built into VSD make it the most obvious candidate among existing DfV methodologies for this type of incorporation. One VSD mainstay is that it should seamlessly integrate into the specific domain that adopts it. This means there is a higher probability that firms and designers will adopt VSD. Similarly, the vast swath of literature that explicates its potency in directing design towards desirable ends makes it particularly apt for adoption into new modes of thought. It can extend the concept of members of stakeholder communities to the broader biosphere, going beyond an artificially constructed, isolated, 'human' stakeholder.

In promoting reflexivity and situatedness, this paper does not and cannot provide a full account of how VSD or DfV approaches more generally can be reconstructed *in toto*. Instead, I have sought to reconceptualise what design means. I have shown that design, although recently marked by philosophical/ethical investigations, remains unrepentantly anthropocentric. The second part of this paper has detailed how posthumanist discourse provides strong arguments against deeply rooted, socio-cultural notions of humanism and anthropocentrism. The analysis further highlights how the 'human-in-isolation' is an illusory concept; nuanced, contingent, and complex interconnections constitute the biosphere, reflexively co-constituting every constituent agent as a result. Design can no longer be relegated to a small corner of this domain of interconnection. Design must adopt an ecological shift in consciousness should it wish to remain salient over time.

To that end, the paper has provided some preliminary guidelines to frame design projects with an ecological mindset. The constraints and efficacy of this proposal are yet to be determined and may be informative. The final section below calls attention to some of its limitations and proposes some potential avenues for future research projects that may prove fruitful towards the promotion of what can be called the (yet to occur) *ecological turn in design*.

## 6. Limitations and Future Research

Over the last thirty years, DfV methodologies in design have been widely discussed within RRI discourse. The turn towards technology in STS and its emphasis on the politics inherent to designed artefacts have initiated investigations into the roles that various actors play during various design stages of technology. As a consequence, investigations extend to how values in technology can be directed towards intended outcomes.

These design methodologies have investigated the impact of various stakeholder groups on what values are designed into or out of technologies as well as how those values are expressed over time. Methodologies rightfully underscore the values of designers and how they can implicitly embed specific values in design [8]. However, they also look at the impacted parties who come into relations (both directly and indirectly) with those products. Unfortunately, implicated values and enrolled stakeholders are limited to humans only. This necessarily relegates any values concerning the environment or nonhuman animals to ancillary levels, along with how the effect on those agents may impact humans either positively or negatively. The primary concern of this paper is the background placement of nonhuman agents alongside how to bring them to the foreground of design consciousness.

This paper argues that an extension of what constitutes stakeholders in DfV approaches is required—not as a second-order consideration for humans, but for those stakeholders per se. I propose several framing tools to reconceptualise how to approach design and create more inclusive design spaces. Such spaces would acknowledge the plurality of phenomenologies that are implicated and existent alongside human agents. Similarly, I propose the theory of contractualism as a preliminary, ad hoc heuristic that can be adopted by designers today as a way to begin this re-envisioning. What is needed still is understanding how to undertake conceptual investigations that formalise an expanded contractualism towards enrolling nonhuman stakeholders. An initial suggestion is the introduction of trustees for these marginalised groups to design teams; however, practical research must be conducted to determine the long-term effectiveness of such a route. Determining if such a strategy can take hold a priori is contentious.

Additionally, the greatest need in DfV research may be the inclusion of a Franco-German (continental) tradition of philosophical inquiry. As things stand, the consultation of philosophical literature in conceptual investigations (VSD in particular) is restricted to the Anglo-American tradition of analytic philosophy. As this paper argues, the most salient discussions of the phenomenologies of nonhuman animals and ecology, in general, have emerged presumably from the former. The Franco-German philosophical tradition proposes strong arguments for ecological thought as ontic. This warrants its inclusion in design, which has substantial material implications for the biosphere at large.

Finally, and following the previous point, DfV and future approaches to design must investigate how objects (entities) come into symbioses with each other. It is apparent, *contra* Latour, that there are asymmetric relations between objects. Many of these serve as event horizons, i.e., points of no return. This means some relations change their constituent objects irrevocably. A self-evident example is anthropogenic climate change and mass diversity loss. Design must account for ways to anticipate symbioses and their effects on entities. In light of this re-imagining of ontological relatedness, anticipatory design is by no means evident. Future research strategies need to explore how symbioses are formed, how technology limits and constrains these relations, and how desirable futures can be realised despite some irrevocable changes.

## 7. Conclusions

Designing for values has made a dramatic impact on the motivations behind technological artefacts and how those artefacts implicate values. The need to direct technologies towards mutually desirable futures has become most overt. Design trends have concurrently realised that the varied nature-cultures of different groups push moral intuitions and design approaches to their limits [10]. This issue is exacerbated by the advent of the transformative nano-bio-info-cogno technologies that implicate and overload moral values. The issue is further intensified when considering the inexorable impact of human innovation on the biosphere.

This paper is relatively unique as it extends a contemporary continental tradition of ecological thinking to the analytical domain of DfV. The recent inauguration "Welcome to the Anthropocene" is arresting as it implicates the weighty impacts of the human quotidian on the once-static background of 'nature' [117]. The sixth mass extinction event makes ignoring the impact of designing on the biosphere unsupportable. Design must take up ecological thought if it is to remain ontologically authentic and relevant; doing otherwise (i.e., continuing as per course) has naturally recalcitrant consequences. This paper has proposed an exploratory set of design principles that can help designers frame design goals and direct design flows towards an ecologically conscious end. Similarly, a contractualist theory of morality—one that can include nonhumans—can provide a provisional heuristic for adoption by analytic designers to initiate this transformation in consciousness. This paper has foregone concrete holism, which runs contrary to the posthumanist analysis it takes up. However, it nonetheless offers some worthwhile principles that can be used to bring about a much needed *ecological turn in design*.

**Funding:** This research received no external funding.

**Institutional Review Board Statement:** Not applicable.

**Informed Consent Statement:** Not applicable.

**Data Availability Statement:** Not applicable.

**Acknowledgments:** This paper has been adapted from a dissertation accepted and published by York University in Canada. I would like to thank Daniel McArthur for providing helpful comments on this manuscript. Any remaining errors are the reviewers' alone. The view expressed in this paper do not necessarily reflect those of the Institute for Ethics and Emerging technologies.

**Conflicts of Interest:** The author declares no conflict of interest.

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
