# Peer review of "The Ecological Turn in Design: Adopting a Posthumanist Ethics to Inform Value Sensitive Design"

_philosophies, doi:10.3390/philosophies6020029_

Round 1
Reviewer 1 Report
The paper discusses the interesting topic – Design for Values within an ecological context. The ideas presented in the paper are certainly worth publishing. However, the paper has several problems that need to be addressed before it can be published.
Several sentences are poorly constructed and thus they are unclear. These sentences need to be corrected by a native English speaker. Here is a partial list of such sentences (9-12,, 13-14, 23-24, 48-49, 53-55, 72, 82-84,111,123-124, 134-136, 158-168,179-180, 198-201, 234, 281-282,
Some statements are hard to understand, i.e. in the line 245 This literature review has so far provided … to which review the author is referring to?
The line in line 304 Because the philosophical focus of this paper is on the ontology of nonhumans in design seems not connected to the claims in this paragraph.
The claim about moral laws in the line 436-438 is false as the moral tradition, in particular virtue ethics, has deep ancient roots, not post-Enlightenment origins.
Line 533 is - the necessity to transform and shift away - to transform what ?
In line 577 Firstly - do not use it if you do not have a second option coming.
The author uses in one or two place cliché claims (Darwin, Copernicus) that should not be in a scientific paper. Their place in a popular science works, as without explanations they are misleading, or not accurate (line 293).
The structures of paragraphs and sections are not well designed. There is no leading statements, there is no “sight post “statements. The reader may wonder how all this discussion is connected together.
The author refers to the paper as a study or a project. The reader may be confused whether the paper reports a study or is a summary of a project.
The claim in line 735 similarly, I proposed the philosophical theory of contractualism as a preliminary, ad hoc heuristic that can be adopted by designers today as a way to begin this re-envisioning. It is hard to find in the paper cogently articulated any the philosophical theory. Many things are said but not theory has been formulated.
The paper seems as a combination of paragraphs selected from the larger document, thus not connected together.
Author Response
Several sentences are poorly constructed and thus they are unclear. These sentences need to be corrected by a native English speaker. Here is a partial list of such sentences (9-12,, 13-14, 23-24, 48-49, 53-55, 72, 82-84,111,123-124, 134-136, 158-168,179-180, 198-201, 234, 281-282,
All of this has been corrected. The entire paper has been professionally copy-edited.
Some statements are hard to understand, i.e. in the line 245 This literature review has so far provided … to which review the author is referring to?
This has now been corrected.
The line in line 304 Because the philosophical focus of this paper is on the ontology of nonhumans in design seems not connected to the claims in this paragraph.
It is connected, both based on the previous statement about the reclamation of animality, as well as the birth of Italian posthumanism as a consequence of animal ethics (the sentence that follows). I have nonetheless clarified this sentence.
The claim about moral laws in the line 436-438 is false as the moral tradition, in particular virtue ethics, has deep ancient roots, not post-Enlightenment origins.
Virtue ethics is not a moral law theory, in fact, it is a good alternative to them. I am careful not to mention virtue ethics in the parentheses of examples of enlightenment moral law theories like utilitarianism and deontology. I have however made this now clear in footnote 12.
Line 533 is - the necessity to transform and shift away - to transform what ?
This has been removed.
In line 577 Firstly - do not use it if you do not have a second option coming.
Done
The author uses in one or two place cliché claims (Darwin, Copernicus) that should not be in a scientific paper. Their place in a popular science works, as without explanations they are misleading, or not accurate (line 293).
Done.
The structures of paragraphs and sections are not well designed. There is no leading statements, there is no “sight post “statements. The reader may wonder how all this discussion is connected together.
Thank you, this is a good suggestion. I have now tried to tie things in a bit more by making the transition between sentences more seamless.
The author refers to the paper as a study or a project. The reader may be confused whether the paper reports a study or is a summary of a project.
Corrected to ‘paper’ throughout.
The claim in line 735 similarly, I proposed the philosophical theory of contractualism as a preliminary, ad hoc heuristic that can be adopted by designers today as a way to begin this re-envisioning. It is hard to find in the paper cogently articulated any the philosophical theory. Many things are said but not theory has been formulated.
I have clarified this here now, as well as throughout exactly what is being done. Thank you for this comment.
Reviewer 2 Report
Overall it was valuable and timely reading a paper on this topic, DfV and VSD. Furthermore, and speaking from the perspective of design practice, the content of this paper is highly relevant and welcomed, even if at times challenging to follow for experts with less than extensive backgrounds in philosophy.
The idea of exploring and advancing post-Anthropocene / VSD design (anticipatory) methods and their basis in a broader philosophical thought (much of it continental) brings rigor to fledgling attempts to envision design methods in the midst of human made - ecological crisis (the 6th mass extinction). The paper has depth even if some of the language might be at times esoteric. Needless to say it was intelligible even if the connectivity between the sections was not always clear. I wondered if it might be valuable to try to connect the thesis of this work more explicitly with an applied architecture, industrial design and art practitioner audience. Perhaps another manuscript could be published in a design-focused journal.
It might also be helpful to connect VSD to other more commonly practiced applied design methods such as Human Centered Design (HCD) to bridge the goals of this work with a larger community of current design practice. Human Centered Design (HCD or Design thinking) is implicitly anthropocentric and very widespread today. Within the design community, designers have proposed adding a fourth albeit incremental dimension to HCD’s Desirable, Feasible, Viable triage: Responsible. ‘Responsible’ is meant to be a catch-all dimension that would demand that design consider impacts of design unintended or not: pollution, killing of animals, climate change etc. Some have suggested a Bio-centric Design (BCD to replace HCD). It is a bit ad-hoc and half-assed but these efforts come from the same spirat and impulse, I believe as the authors work here. Again, this work could help influence the design community with more direct discourse and dialogue.
Other than a few spelling and wording suggestions to increase readability, I would also suggest shortening the philosophical literature review if possible. It is of course very valuable to situate this work in a larger intellectual context but perhaps the discussion’s overall thread gets a bit overwhelmed by that section’s length. Another editorial pass of the paper just for reading clarity would make the paper even stronger.
Specific Comments:
Line 205 Are you referring to Human Centered Design processes or are you avoiding this classification?
Line 225 Is HCI Human- Computer Interaction?
Line 247 Timothy Morton calls the 'ecological thought'
Line 308 flavor or posthumanistm
Line 469 ontological landscape which can then be built upon...
Line 471 the structure in which the posthuman can grow....
Line 524 architecture (do you mean design? it is broader than archtitecture)
Line 587 I would love to see more examples of design and technical design that demonstrate your points like this.
Line 742 "need most in DfV research is the inclusion of the Franco-German (continental) traditional of philosophical inquiry (in addition to Anglo American analytical philosophy.... yes indeed!
Author Response
The idea of exploring and advancing post-Anthropocene / VSD design (anticipatory) methods and their basis in a broader philosophical thought (much of it continental) brings rigor to fledgling attempts to envision design methods in the midst of human made - ecological crisis (the 6th mass extinction). The paper has depth even if some of the language might be at times esoteric. Needless to say it was intelligible even if the connectivity between the sections was not always clear. I wondered if it might be valuable to try to connect the thesis of this work more explicitly with an applied architecture, industrial design and art practitioner audience. Perhaps another manuscript could be published in a design-focused journal.
Actually, I have intentions of doing exactly this in another paper. If the reviewer is interested, after the publication of this manuscript they could contact me and we can work together on a paper that does exactly this!
It might also be helpful to connect VSD to other more commonly practiced applied design methods such as Human Centered Design (HCD) to bridge the goals of this work with a larger community of current design practice. Human Centered Design (HCD or Design thinking) is implicitly anthropocentric and very widespread today. Within the design community, designers have proposed adding a fourth albeit incremental dimension to HCD’s Desirable, Feasible, Viable triage: Responsible. ‘Responsible’ is meant to be a catch-all dimension that would demand that design consider impacts of design unintended or not: pollution, killing of animals, climate change etc. Some have suggested a Bio-centric Design (BCD to replace HCD). It is a bit ad-hoc and half-assed but these efforts come from the same spirat and impulse, I believe as the authors work here. Again, this work could help influence the design community with more direct discourse and dialogue.
VSD is actually interesting because it is not exclusive of other methodologies, although it is a methodology proper. I have made modifications to the section explicitly on VSD as to how it relates to parallel methodologies. I have used what you have written in the footnotes in that section to make it more convincing. Thank you for the comment.
Other than a few spelling and wording suggestions to increase readability, I would also suggest shortening the philosophical literature review if possible. It is of course very valuable to situate this work in a larger intellectual context but perhaps the discussion’s overall thread gets a bit overwhelmed by that section’s length. Another editorial pass of the paper just for reading clarity would make the paper even stronger.
Yes, I prefer not to remove the literature because, as you said, I want this paper also to be a practice in situating. However, the extensive copy-editing that I have commissioned has certainly increased the readability of the paper significantly.
Specific Comments:
Line 205 Are you referring to Human Centered Design processes or are you avoiding this classification?
This clarification, as per your previous comment, is made in the introductory section on VSD as well as in footnote 2.
Line 225 Is HCI Human- Computer Interaction?
Yes, I have made sure to clarify that on the first use now (line 125).
Line 247 Timothy Morton calls the 'ecological thought'
Clarified.
Line 308 flavor or posthumanistm
Corrected.
Line 469 ontological landscape which can then be built upon...
Corrected.
Line 471 the structure in which the posthuman can grow....
Corrected.
Line 524 architecture (do you mean design? it is broader than architecture)
No here I meant architecture specifically given that it is particularly part of the Italian posthumanist project and they are very clear in using that work rather than the broader word ‘design’.
Line 587 I would love to see more examples of design and technical design that demonstrate your points like this.
Done!
Line 742 "need most in DfV research is the inclusion of the Franco-German (continental) traditional of philosophical inquiry (in addition to Anglo American analytical philosophy.... yes indeed!
I agree! I hope that I have done at least a small step here towards doing that. As mentioned, if you are interested in exploring this please do let me know post-publication!
Round 2
Reviewer 1 Report
No further comments. My concerns have been addressed
This manuscript is a resubmission of an earlier submission. The following is a list of the peer review reports and author responses from that submission.
Round 1
Reviewer 1 Report
The paper opens an interesting discussion. However, it needs extensive editing of grammar and structure. The argument is not very clearly articulated. The objectives are not not clear. Terms are not clearly defined. e.g., can we port human values into the post humanism without chaging their content, what would the post humanist ethics encompassing all living things contain. The paper would benefit from more focused presentation of the main idea ( which is what ?) rather than from the discussion of several side issues.
Final comment: good ideas, good topic, needs more cohesive account
Author Response
The paper has been heavily edited to make parts clearer. Similarly, I have worked through the paper to make sure the language has a stricter logical flow.
Reviewer 2 Report
I have made comments using line notes from the manuscript in the attached file. Overall the article is fine, but there are things that need clarification or development

Author Response
44 progressivism has naturally lead to an inability to discretely separate any technology from its
45 situatedness
Most? I’m not sure the case has been made to say all nor am I sure it is essential to the overall argument.
clarified
47 social. A consequence of this is that a constrained landscape of potential futures is open to us, one
48 that is delimited by this co-construction.
Which is delimited? The landscape or the potential futures? Needs clarifying
clarified
48 How we design technologies then, based on how they can
49 constrain our choices about the future, becomes a valuable question in the twenty-first century
I don’t disagree but how does this follow from the preceding text? Should how we choose design be based on how choices are constrained? Again clarification is needed here
clarified
62 to help incorporate human values in early design phases. This Design for Values (DfV) typically come
“This Design for Values” what? It seems that a term is missing, approach, method? It could affect whether come is singular or plural
yes, fixed
68 STS use first then abbreviate
done
72 strike this before anthropocentric
done
81 all DfV approaches or just VSD?
all, clarified
98 section 2 sentence seems a run on
fixed, divided into two.
104 It is confusing to list the sections as 1-4 above and then section 1 is now 2
numbering fixed
124 ICT use then abbreviate
done
126 why only Worth Centered Computing capitalized? Consistency
done
127 some scholars – who
added
130 strike an before the quotes
done
261 in this dissertation – edit text
done
419-420 Are you suggesting progress or teleology in evolution? if so this will need more support
against teleolgy. Clarified.
37-439 it sounds like body-oriented ethics is deontological. Clarification is needed to distinguish them
clarified in footnote.\
583 last word too or to?
fixed (to)
597-601 How are these related? They do not seem to directly follow from the other
clarified
621 Do you support contractualism? If so, it should be mentioned how you will adjust it to fit your scheme in this section, if not, it seems not terribly relevant to the overall discussion to give so much info on it and then dismiss it
643 Is this why you are giving attention to contractualism?
This is clarified now at the end of the preceding section to frame the intent of this section.
653 maybe we-word. Why intuitively? All adequate moral theories must… 653-663 whole paragraph could use more development, especially if you want to appeal to contractualism or the concept of trustees for nonhuman nature
modified
654 Michael Walzer argues that membership is the most important and most contested aspect of contract/community.
hence the point of this section and clarified in the research limitations section that proceeds it.
I can’t tell at the end of page 15 whether you want to accept a modified contractualism or whether you reject it? If rejected completely, it seems to be a page or two that is unnecessary and if you want to use it, that needs clarified about what you are bringing forward in the argument 728-729 it sounds like you do want to carry contractualism forward
this has been clarified, as per above and in the section itself.